# Distinct Emotional and Cardiac Responses to Audio Erotica between Genders

**DOI:** 10.3390/bs13030273

**Published:** 2023-03-20

**Authors:** Zhongming Gao, Xi Luo, Xianwei Che

**Affiliations:** 1Department of Neurology, The Affiliated Hospital of Hangzhou Normal University, Hangzhou 310000, China; 2College of Preschool and Primary Education, China West Normal University, Nanchong 637000, China; 3Centre for Cognition and Brain Disorders, The Affiliated Hospital of Hangzhou Normal University, Hangzhou 310000, China; 4TMS Centre, Deqing Hospital of Hangzhou Normal University, Hangzhou 313200, China

**Keywords:** gender difference, erotica, emotions, heart rate

## Abstract

Emotional and cardiac responses to audio erotica and their gender differences are relatively unclear in the study of the human sexual response. The current study was designed to investigate gender differences regarding positive and negative emotional responses to erotica, as well as its association with cardiac response. A total of 40 healthy participants (20 women) were exposed to erotic, neutral, and happy audio segments during which emotions and heart rate changes were evaluated. Our data showed distinct emotional responses to erotica between genders, in which women reported a higher level of shame than men and rated erotic audios as less pleasant than happy audios. Meanwhile, men reported erotic and happy audios as equally pleasant. These results were independent of cardiac changes, as both sexes demonstrated comparable heart rate deceleration when exposed to erotica relative to neutral and happy stimuli. Our results highlight the role of sociocultural modulation in the emotional response to erotica.

## 1. Introduction

Sexual arousal is a key component of the human sexual response, which has been extensively investigated for decades [1,2,3,4]. It is generally accepted that men demonstrate higher sexual arousal and show higher concordance between physiological and subjective arousal than do women [5,6]. However, the human sexual response is a dynamic combination of physiological, cognitive, and emotional processes, of which emotional responses to erotica and its gender differences are relatively unclear.

Erotica (or erotic stimulus) is usually known as a quality (or material) that can cause sexual desire or arousal. Exposure to erotica is associated with a range of positive and negative emotional experiences, ranging from pleasant, passionate, ashamed/embarrassed, and disgusted/aversive [7]. Among positive emotions, pleasure has been identified as the principal variance in emotional meaning [8,9], and it has been the most widely used emotion in the study of emotional responses to erotica [7,10]. Indeed, erotica is believed to be associated with increased pleasure, as it is characterised by a high level of appetitive motivation and arousal [7,10,11]. Moreover, men tend to rate erotic stimuli as more pleasant than do women [12,13]. Interestingly, erotic stimuli can also provoke unique negative emotions, especially in women, with shame or embarrassment being most commonly reported [12,14,15]. This set of negative emotions is suggested to reflect normative expectations that women tend to report altered sex behaviours or experiences to meet these expectations [16]. Overall, there seems to be a gender difference in emotional patterns in the context of erotica. To date, there is no evidence to directly compare the emotional properties induced by erotic stimuli in men and women. One study demonstrated that a higher proportion of women identified embarrassment, while men reported excitement in response to erotic pictures [12].

It is noted that previous studies have predominantly presented visual erotic stimuli in the studies of sexual response [12,13]. Audio erotica has been scarcely evaluated in the context of sexual arousal or erotic emotions. Only one study presented audio narrative erotica and demonstrated similar genital responses to those induced by visual or visual-audio stimuli depicting couples engaged in sexual activities [17]. Specifically, both sexes demonstrated significant increases in genital response to audio erotica, with men being more selective to preferred materials [17]. Audio erotica is a critical component of sexual arousal and sex experience. However, our understanding of sexual arousal and emotional responses to audio erotica is highly limited.

It is also interesting to know determine whether different emotional responses to erotica between genders are characterized by physiological changes. It is well established that exposure to erotica is associated with cardiac deceleration in the early stage, indicative of orienting and sensory intake, followed by cardiac acceleration supporting defensive action [10,12]. One study also examined gender differences in cardiac response to erotica, with results depending on the type of stimuli. Specifically, both sexes were comparable in cardiac deceleration as well as acceleration when exposed to pictures of erotic couples, but opposite-sex erotica was associated with larger cardiac deceleration in women [12]. However, it is noted that these studies employed visual erotic stimuli [10,12]; cardiac response to audio erotica has not been reported, nor have gender differences. Compared to visual stimuli, erotic audios, such as moaning and thrusting sounds, represent a dynamic content of sexual experiences and may garner at least comparable or even much more attention than visual stimuli. Exposure to erotic audios is therefore expected to be associated with cardiac deceleration to support attention orientation [7].

The current study was designed to investigate gender differences regarding positive and negative emotions in response to erotica, as well as the association with cardiac response. Previous studies have extensively investigated physiological arousal in response to erotica [18,19]. Meanwhile, the current study focused on the emotional properties of erotica, as well as the differences in gender responses, which would enrich our understanding of the human sexual response. Whereas visual erotica has been predominantly evaluated in the literature, this study uniquely presented audio erotica, as it is a critical component of sexual arousal and sexual experience. It is noted that participants from Western cultures were consistently recruited in previous studies. This study recruited participants from an Eastern culture, the results of which would add to the understanding of emotional response to sexual arousal across cultures. In addition, we assessed heart rate changes to evaluate physiological sexual arousal, findings from which would shed light on the possible reasons for distinct emotions between genders in response to erotica.

In the current investigation, healthy male and female participants were exposed to erotic, neutral, and happy audio segments, while they reported their level of pleasure and shame. Happy stimuli were included as an active control condition. An electrocardiogram (ECG) was used to measure heart rate changes to the audio stimuli. We hypothesized that male and female participants would be associated with higher pleasure and shame emotions, respectively. It was also assumed that both genders would demonstrate comparable heart rate changes, as we used stimuli of erotic couples (see [12]).

## 2. Methods

### 2.1. Participants

Sample size calculation was performed to determine the minimum sample size needed to power a mixed ANOVA design [20]. Specifically, the significance level (alpha) and power were set to 0.05 and 0.8, respectively. In order to achieve a medium effect size (Cohen’s d = 0.5), a total sample of 32 was needed, which determined a critical F and $\eta_{p}^{2}$ value of 4.15 and 0.2, respectively. A group of 40 adults were therefore recruited in this study (age range: 18–27 years, mean ± SD = 19.25 ± 1.81, all Han Chinese, 20 women). The determined F and $\eta_{p}^{2}$ values were comparable to those reported in the main findings. It is noted that ECG data were unavailable, due to technical issues, for two females and one male.

Potential participants were recruited through flyers posted in libraries in the China West Normal University. All the participants were heterosexual by means of self-report (‘heterosexual’, ‘homosexual’, ‘bisexual’, ‘gender non-conforming’, ‘transgender’, ‘prefer not to indicate’). In order to reduce expectancy effects, participants were told that the aim of the study was to examine heartbeat response to audio fragments. Exclusion criteria included use of psychoactive medication or a history or current diagnosis of a psychiatric disorder, as assessed by the Mini International Neuropsychiatric Interview (MINI) [21]. Being pregnant or currently experiencing menstrual bleeding were also exclusion criteria, as pregnancy tends to alter sexual desire [22], and menstrual phases can modulate sexual responses to erotica [23]. All study participants provided informed consent and the experiment was approved by the Ethics Committee in the China West Normal University. This study was conducted in Sichuan, China, in accordance with the Declaration of Helsinki.

### 2.2. Experimental Design and Procedure

A single-session, mixed design protocol was used in this study. Gender was the between-subject variable, while the within-subject variable was stimuli that included erotic, neutral, and happy audio fragments.

Following providing consent, participants were set up with the ECG recording system (see below in Section 2.5) and asked to adjust the volume on the earphones. Participants were then provided with the audio tasks (see below in Section 2.3) in which they rated the level of pleasure and shame produced by the audio fragments.

### 2.3. Experimental Protocol

The participants underwent 3 blocks of audio tasks, each including 6 trials. Each block started with a fixation cross for 2 s, which was followed by the presentation of a 5 s audio fragment (2 in each category, 6 in total) (Figure 1). The participants were then asked to rate the level of “pleasure” (‘How much do you feel pleased by the audio’, as used in [10]) and “shame” (‘How much do you feel ashamed by the audio’, as used in [15]) in 4 s of each audio fragment on a scale of 0 to 10 (0–10: not at all to extremely intense). The ratings were provided on a piece of paper that could not be seen by the experimenter. Each trial ended with a white noise segment lasting for 3 s to control carryover effects. The 6 trials included in a certain block were randomized in such a way that 2 trials of the same category (e.g., erotic) were not presented consecutively. Participants took a break for 5 min between blocks. The experimenter was outside of the lab during the audio task to control social judgements.

### 2.4. Experimental Stimuli

Another group of 19 participants was initially recruited to evaluate the experimental stimuli used in this study (age range: 18–33 years, mean ± SD = 19.64 ± 2.32, all Han Chinese, 9 women). The same inclusion and exclusion criteria as those used above were used for this group (see *Participants*). The experimental protocol was same as that used in the previous audio tasks, except that the participants were asked to rate the “valence” and “arousal” of the audio fragments. Valence describes the extent to which an emotion is positive or negative (0–10: extremely negative to extremely positive, ‘Please indicate the valence of the audio’), whereas arousal refers to the strength of the associated emotional state (0–10: no arousal to extreme arousal, ‘Please indicate the level of arousal of the audio’) [24]. A total of 30 audio fragments (5 blocks, 10 in each category) were extracted from the Internet. The erotic, neutral, and happy audio fragments were extracted, respectively, from free sites (https://www.pornhub.com/ (accessed on 4 October 2019); https://www.ximalaya.com/jiaoyupeixun/2808888/ (accessed on 5 October 2019); https://sc.chinaz.com/tag_yinxiao/XiaoSheng.html (accessed on 5 October 2019)). Specifically, the erotic audios were extracted, which included sounds of having sex, such as moaning (mainly from women), thrusting, and sounds of water and lubrication. The neutral audios were extracted from sounds of Mandarin reading, such as the introduction of stones, technical reports, and weather forecasts. Meanwhile, the happy audios were of laughter.

Chi-square tests based on the arousal data were used to select the experimental stimuli (see Appendix A). Specifically, *p* values of Chi-square tests were ranked for each stimulus. Six stimuli which showed the largest *p* values between genders were selected in each category. As a result, male and female participants were matched according to both arousal and valence for a certain stimulus. We also compared the selected stimuli between the categories. Specifically, two-way mixed ANOVAs (gender X stimuli) and post-hoc pairwise comparisons revealed that the selected erotic and happy stimuli were comparable in both arousal (*P_Bonf_* = 0.99) and valence (*P_Bonf_* = 0.99). As expected, both the erotic (*P_Bonf_* = 0.013, *M* = 5.60, *SD* = 1.95) and happy stimuli (*P_Bonf_* = 0.014, *M* = 5.31, *SD* = 2.29) showed higher arousal levels than the neutral stimuli (*M* = 3.96, *SD* = 2.28). The selected stimuli were also comparable in valence (*P_Bonf_* = 0.99). We also tested the correlation between the valence and arousal in the selected stimuli, and the results showed a significant positive correlation across stimulus categories (*r* = 0.43, *p* = 0.001).

### 2.5. ECG Recordings

ECG was recorded using a BITalino (r)evolution Board Kit BT (PLUX Biosignals, Lisbon, Portugal) (http://bitalino.com/en/, accessed on 16 February 2023). Three disposable Ag/AgCl electrodes with electrolyte gel were used with Velcro, with two being attached to the bilateral clavicle area within the rib cage and one electrode to the lower edge of left rib cage, respectively. Data were recorded with OpenSignals (r)evolution software (v.2017, PLUX Biosignals, Lisbon, Portugal) in the sampling rate of 1000 Hz.

### 2.6. Data Analysis

For ratings of pleasure and shame, data were averaged across trials for each participant. For the ECG data, inter-beat-interval (IBI) series were derived by the Pan–Tompkins algorithm that identifies the peak of the R wave as the fiducial point [25,26,27]. Artefacts were visually checked and edited if necessary, according to published guidelines [28]. IBI series were then transformed to beat-per-minute (BPM) series and baseline corrected for each trial, in which a 2 s fixation baseline (−2–0 s, where time 0 represents the onset of the audio clip) was subtracted from the whole trial containing the phases of image presentation and ratings. This method is believed to control for individual differences in baseline heart rate and to capture the dynamics of event-related heart rate change in a short period [10,29]. Heart rate data were then averaged across trials for each participant, and the area under the curve (AUC) was calculated with the linear trapezoidal rule to measure event-related heart rate change during the presentation and evaluation of audio segments. For two females and one male participant, heart rate recordings were unavailable due to technical issues.

### 2.7. Statistical Analyses

Using SPSS (version 23; IBM Corp, Armonk, NY, USA), mixed two-way ANOVAs were performed to examine the main and interaction effects of gender (men, women) and stimuli (erotic, neutral, and happy) on ratings of pleasure and shame, as well as on the AUC of heart rate changes. Post hoc pairwise comparisons were then conducted to explore the significant main and interaction effects, with the Bonferroni correction and the α-level set to 0.05. It is noted that we divided the original alpha level (0.05) by the number of comparisons in the Bonferroni corrections (e.g., 0.05/3 in the comparisons of a 3-level stimuli). We also performed a series of bivariate correlation analyses comparing cardiac changes and pleasure as well as shame, both across genders and in each gender group.

### 2.8. Supplementary Analysis

As both genders appeared to demonstrate different dynamics in heart rate changes (i.e., relative to baseline) when exposed to erotic segments, we further performed a mixed two-way ANOVA on heart rate changes. Specifically, gender (men, women) and time (baseline, image presentation, during pleasure rating, during shame rating) were specified as the between-group and within-group factor, respectively. The same method was also applied to the neutral and happy stimuli.

## 3. Results

### 3.1. Pleasure Ratings

As shown in Figure 2A and Table 1, the two-way ANOVA revealed a significant gender × stimuli interaction effect on pleasure ratings (F_2,76_ = 4.13, *p* = 0.026, $\eta_{p}^{2}=0.098$). Post hoc pairwise comparisons indicated that women reported a higher pleasure response to happy stimuli (*M* = 5.55, *SD* = 2.57) compared to erotic (*P_Bonf_* = 0.007, *M* = 2.86, *SD* = 2.67) and neutral stimuli (*P_Bonf_* = 0.013, *M* = 3.29, *SD* = 2.58). Meanwhile, male participants reported a higher pleasure response to happy (*P_Bonf_* = 0.016, *M* = 4.47, *SD* = 2.13) and erotic (*P_Bonf_* = 0.016, *M* = 4.43, *SD* = 1.73) stimuli compared to neutral stimuli (*M* = 2.43, *SD* = 2.05).

### 3.2. Shame Ratings

The two-way ANOVA revealed a significant gender × stimuli interaction effect on shame ratings (F_2_,_76_ = 9.08, *p* = 0.003, $\eta_{p}^{2}=0.19$) (Figure 2B and Table 1). Post hoc pairwise comparisons showed that erotic stimuli (women: *M* = 6.66, *SD* = 2.27, men: *M* = 4.59, *SD* = 2.44) induced higher levels of shame compared to neutral (women: *P_Bonf_* = 0.001, *M* = 0.09, *SD* = 0.27, men: *P_Bonf_* = 0.001, *M* = 0.27, *SD* = 0.65) and happy (women: *P_Bonf_* = 0.001, *M* = 0.32, *SD* = 0.78, men: *P_Bonf_* = 0.001, *M* = 0.73, *SD* = 1.48) stimuli in both genders. Moreover, women (*M* = 6.66, *SD* = 2.27) reported higher shame compared to men (*M* = 4.59, *SD* = 2.44) in response to erotic stimuli (*P_Bonf_* = 0.01).

### 3.3. Heart Rate Changes

Figure 3A demonstrated the dynamic heart rate deceleration induced by different stimuli. For the image presentation stage, a two-way ANOVA revealed the main effect of stimuli on heart rate change (F_2,70_ = 7.66, *p* = 0.002, $\eta_{p}^{2}=0.18$) (Figure 3B and Table 1). Post hoc pairwise comparisons indicated that erotic stimuli (*M* = −2.13, *SD* = 2.31) induced a larger heart rate deceleration compared to neutral (*P_Bonf_* = 0.016, *M* = −0.93, *SD* = 1.89) and happy stimuli (*P_Bonf_* = 0.007, *M *= −0.44, *SD* = 1.76), regardless of gender. Heart rate changes were not associated with the level of pleasure or shame (all *p* > 0.05).

During pleasure rating, a two-way ANOVA revealed a main effect of stimuli on heart rate change (F_2,70_ = 18.43, *p* = 0.000, $\eta_{p}^{2}=0.35$) (Figure 3C and Table 1). Post hoc pairwise comparisons indicated that erotic stimuli (*M* = −2.51, *SD* = 3.53) induced a larger heart rate deceleration compared to neutral (*P_Bonf_* = 0.003, *M* = −0.06, *SD* = 2.92) and happy stimuli (*P_Bonf_* = 0.000, *M *= 1.40, *SD* = 2.31), regardless of gender. Happy stimuli (*M* = 1.40, *SD* = 2.31) also induced increased heart rate compared to neutral stimuli (*P_Bonf_* = 0.035, *M *= −0.06, *SD* = 2.92).

During shame rating, a two-way ANOVA also revealed the main effect of stimuli on heart rate change (F_2,70_ = 18.10, *p* = 0.000, $\eta_{p}^{2}=0.34$) (Figure 3D and Table 1). Post hoc pairwise comparisons indicated that erotic stimuli (*M* = −1.15, *SD* = 3.16) induced a larger heart rate deceleration compared to neutral (*P_Bonf_* = 0.011, *M* = 0.89, *SD* = 2.43) and happy stimuli (*P_Bonf_* = 0.000, *M *= 2.56, *SD *= 2.55), regardless of gender. Happy stimuli (*M* = 2.56, *SD *= 2.55) also induced increased heart rate compared to neutral stimuli (*P_Bonf_* = 0.024, *M* = 0.89, *SD* = 2.43).

### 3.4. Supplementary Results

A mixed ANOVA revealed no gender × time interaction effect on the erotic data (F_3,105_ = 0.94, *p* = 0.398, $\eta_{p}^{2}=0.03$), suggesting that both genders were comparable in heart rate deceleration when exposed to erotic stimuli. There was only a time effect (F_3,105_ = 12.95, *p* = 0.000, $\eta_{p}^{2}=0.27$), which indicated that heart rate was decelerating across genders.

## 4. Discussion

The current study was designed to investigate gender differences in the emotional response to erotica. Audio erotica, which is a key component of human sexual arousal and sexual experience, was uniquely presented. Our results demonstrated distinct emotions regarding erotica between genders, with women reporting a higher level of shame compared to men, and rating erotic stimuli as neutral. Meanwhile, men tended to feel more pleasant when exposed to erotic relative to neutral stimuli. Cardiac data indicated that both genders showed comparable heart rate deceleration in response to erotica compared to neutral and happy stimuli.

Compared to men, women reported a higher level of shame in the context of erotica (Figure 2B). Previous studies indicated that women tend to experience unique negative emotions in response to erotic stimuli, such as shame and embarrassment [12,14,15]. Our findings extended these results by demonstrating that both sexes could be ashamed of erotica, and more interestingly, that women tend to feel more ashamed than men. Possible reasons for this distinction are discussed later.

In terms of pleasure, women reported erotic stimuli to be equally pleasant as compared to neutral ones (Figure 2A). This is consistent with the finding that women tended to rate erotic pictures as neutral [13]. However, contrary to our hypothesis, in our data, men did not report higher pleasure to erotic stimuli than women. The literature indicated that men felt more pleasant when exposed to erotica compared to women [10,13]. Although our result suggested that both sexes were different in reported pleasure to some extent (*M_men_* = 4.43, *M_women_* = 2.86), it did not reach a level of significance. It is possible that male participants did not favour the erotic audios used in this study, and therefore, they reported medium levels of pleasure.

More interestingly, men reported erotic stimuli to be equally pleasant compared to as happy stimuli, while women rated happy stimuli to be more pleasant than erotic stimuli. It is widely accepted that men have higher sexual desire [30,31] and sexual arousal [19,32], and thus report erotica as more pleasant [13]. However, the sexual response of women is shaped by contextual (e.g., commitment) and social (culture, beliefs) factors, to a larger extent [31,33]. Our data of the cardiac response may provide further evidence supporting this argument. Specifically, the two sexes showed comparable heart rate deceleration to erotica compared to the control stimuli (Figure 3). Heart rate deceleration is believed to support orienting and sensory intake to external stimuli in the early stage of defence [10,12,34]. Both genders were alert and attentive while listening to erotic sounds. Moreover, heart rate changes were not associated with either pleasure or shame whatsoever. Therefore, both genders may demonstrate equal bodily involvement in the erotic stimuli, suggesting that the distinct emotional patterns between sexes may be a result of other processes.

One possibility is that the sexual response of women is influenced by social desirability and social expectation. Women tend to show lower concordance between increased genital arousal and self-reported sexual arousal [5,6]. Furthermore, gender difference in self-reported sexual behaviours were most prominent in a threat context, whereby the experimenter could potentially view participant’s response, especially for behaviours considered less acceptable for women than men (e.g., masturbation) [16]. Our data indicated that both sexes were equally engaged in the erotic stimuli, but women reported higher levels of shame than men and preferred happy audios more than erotic ones. These findings together highlight the significance of sociocultural factors in shaping a woman’s sexual response. Indeed, it is proposed that gender differences in sexuality may mirror sex roles, and individuals may alter their self-presentations to meet these roles [16,35].

Men and women also differ in neural responses to erotica, which may be associated with distinct emotional responses. Although both sexes showed similar patterns in brain activation to erotica [18,19], men demonstrated increased brain activation in the amygdala than women [36]. Meanwhile, women showed increased brain activity mainly in the cingulate and insular cortex [19]. These findings indicated that men may attribute a greater appetitive incentive value to erotica, while exposed to erotica may be associated with more interoceptive awareness in women.

It is noted that distinct emotional patterns to erotica between sexes may result from a combination of both sociocultural and neural modulation. Emotions are believed to be products of neuronal firing, which is also shaped by social and environmental contexts. It is also important to consider the role of hormones in the association between physiological and emotional responses to erotica in our data. Kisspeptin, a family of peptide hormones cleaved from the product of the Kiss1 gene, is believed to be potential candidate in fulfilling this role. Human studies indicated that kisspeptin signaling may be able to integrate sensory processing with limbic pathways involved in sexual arousal and mood [37,38]. Besides, other endocrine mediators, including oxytocin, cortisol, and vasopressin, are of great importance in integrating physiological reproductive processes with essential emotions [39].

We also presented an interesting cardiac response in the phase of emotional evaluation (Figure 3C,D). Previous studies generally demonstrated cardiac response during the presentation of erotic stimuli [10,12]. Our study extended the time window and showed heart rate acceleration, especially in the evaluation of positive stimuli. This finding may not be associated with movement, as it was unique to positive stimuli. Cardiac acceleration in the rating of positive stimuli is potentially indicative of positive emotions. It is noted that heart rate changes were on the order of a few beats per minute, which did not result from measurement uncertainty. In one way, this magnitude of change is robust across ECG devices, as it is the same (~3) as those found in the anchoring studies in this field by Bradley et al. [10,12]. ECG signals acquired with our device have been demonstrated to be highly consistent with those of other established biomedical toolkits [40,41] (Batista, de Silva, and Fred 2017; Batista, de Silva, Fred, et al., 2019). In another way, the device used in this study is also reliable and accurate in responding to other types of stimuli, such as painful stimuli [25,27].

There are some limitations in the study. Sexual experience is suggested to affect sexual responses [42], but we were not able to examine the influence of sexual experience on our findings, as there were only a few participants (2 women and 3 men) who reported having sexual experience. Other factors, such as religions and core beliefs, may also have an impact on emotional responses to erotica, which need to be evaluated in this context. It also indicates that erotic audios and images may not employ the exact same pathways for sexual arousal. Moreover, the experimental materials were slightly different in terms of verbal content (e.g., sounds of having sex vs. sounds of Mandarin reading vs. laughter), which might be a potential confounder of the findings. Nonetheless, our statistics indicated that the erotic and happy stimuli were comparable in arousal and valence, and that all three types of stimuli were matched in valence. The experimental materials were different in the load of moaning, in which females weighed slightly over males. This could have an impact on the emotional responses to erotica between genders. We have randomly clipped ten audios from online sources, but there seemed to be a tendency that females moan over males in sexual intercourse. Our study therefore calls for special control over the moaning sound between genders in future studies. In addition to moaning, our experimental materials matched other types of sounds in sexual behaviours.

Although the experimenter was outside of the lab in order to control social desirability, we did not provide an objective measure of social desirability (e.g., Marlowe–Crowne Social Desirability Scale) [43]. This could be a potential confounder, as the mere presence of another person could modulate a participant’s response [44,45]. Consistent with previous studies [10,15], self-reported pleasure and shame were evaluated in the current study. Although we have demonstrated findings consistent with the results of these studies, it is acknowledged that self-reported pleasure and shame were based on a single item, which did not allow for the evaluation of the reliability using Cronbach’s alpha. This study was limited to heterosexual participants. It would be highly interesting to extend this study to other groups, such as homosexual, bisexual, gender non-conforming, or transgender participants. Findings from those studies would greatly enrich our understanding of emotional responses to erotica and the association with cardiac responses.

Findings from this study may have implications for future investigations. There is evidence that women demonstrated genital arousal in response to both preferred and nonpreferred gender, while men reported higher sexual arousal to female stimuli [46,47]. Erotic audios used in this study were composed mostly of moaning of women, followed by sounds of men moaning, thrusting, and water. We detected unique emotional responses to erotica in men and women, characterised by more pleasure and shame, respectively. Future studies may wish to further characterise this pattern of emotions regarding the exact contents of erotica, e.g., erotic couples, and same- and opposite-sex erotic stimuli. We evaluated the two most often used emotions (e.g., pleasure and shame), leaving other related emotional responses to be assessed in future studies, such as passion, interest, and aversion/disgust. Findings regarding a range of emotions would provide a broader picture of human emotional response to sexual arousal. It is noted that our participants had an Eastern background, which may be more conservative regarding the expression of sexual emotions. It would be interesting for future studies to directly compare the expression of sexual emotions between cultures. Our results also have implications in understanding gender differences in sexuality for sex therapy, as well as encouraging freedom of sexual expression to reduce risk of sexual disorders for both females and males.

To conclude, we demonstrate distinct patterns of emotional response to audio erotica between genders. Moreover, this effect is independent of cardiac deceleration, indicating the level of engagement in erotica. Our findings may add to our understanding of sexual responses, as well as gender differences, by highlighting distinct emotional responses to audio erotica.

## Figures and Tables

**Figure 1 behavsci-13-00273-f001:**
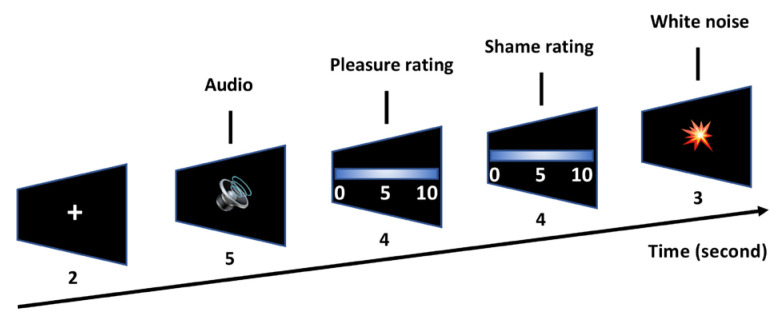
Experimental protocol. A single block is presented here for illustration purposes. Each block consisted of 6 trials, with 2 in each stimulus category. A total of three blocks were performed.

**Figure 2 behavsci-13-00273-f002:**
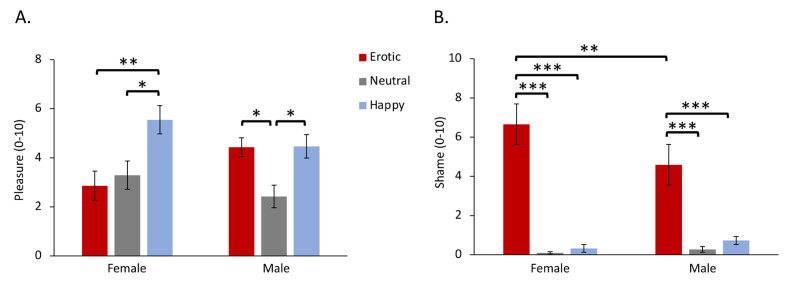
Self-reported pleasure (**A**) and shame (**B**) as a function of gender and stimuli. * *p* ≤ 0.05, ** *p* ≤ 0.01, *** *p* ≤ 0.001.

**Figure 3 behavsci-13-00273-f003:**
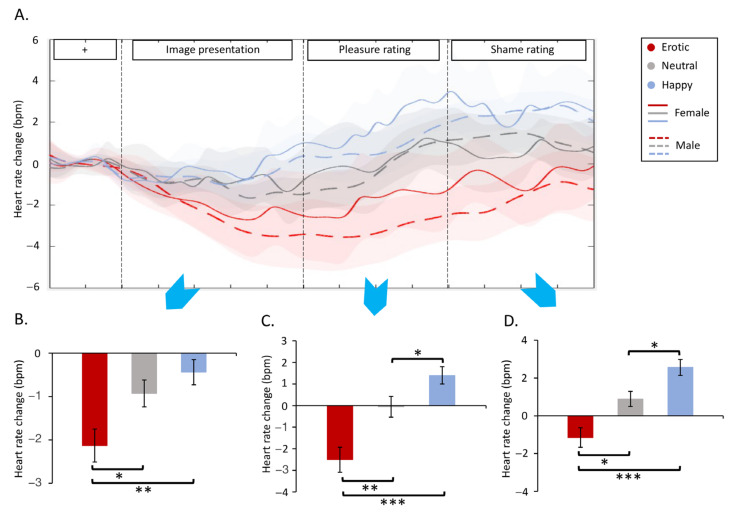
Heart rate results. (**A**) Heart rate dynamics across gender groups and stimuli (**B**–**D**) indicated heart rate statistics in the image presentation, pleasure rating, and shame rating phases, respectively. Dashed lines outline these phases; bpm denotes beat-per-minute; * *p* ≤ 0.05, ** *p* ≤ 0.01, *** *p* ≤ 0.001.

**Table 1 behavsci-13-00273-t001:** Summary of ANOVA results. Both erotic and happy stimuli induced medium pleasure levels in men, but only erotic stimuli did for women. Meanwhile, women reported a higher level of shame compared to men in responding to erotica. In addition, both genders showed comparable heart rate deceleration to erotic stimuli. * *p* ≤ 0.05, ** *p* ≤ 0.01, *** *p* ≤ 0.001.

Source	*df*	Mean Square	*F*	*p*	$\eta_{p}^{2}$
(*Pleasure ratings*)	1	0.47	0.09	0.770	0.00
Gender	2	47.33	8.98	0.000 ***	0.19
Stimuli	2	21.79	4.13	0.026 *	0.10
Gender × Stimuli					
(*Shame ratings*)					
Gender	1	7.33	2.38	0.130	0.06
Stimuli	2	371.59	180.83	0.000 ***	0.83
Gender × Stimuli	2	18.66	9.08	0.003 **	0.19
(*Heart Rate-during image presentation*)					
Gender	1	2.84	0.58	0.450	0.02
Stimuli	2	30.99	7.66	0.002 **	0.18
Gender × Stimuli	2	0.02	0.01	0.990	0.00
(*Heart Rate-during pleasure rating*)					
Gender	1	23.98	2.26	0.142	0.06
Stimuli	2	157.09	18.43	0.000 ***	0.35
Gender × Stimuli	2	1.88	0.22	0.784	0.01
(*Heart Rate-during shame rating*)					
Gender	1	1.29	0.15	0.698	0.00
Stimuli	2	129.05	18.10	0.000 ***	0.34
Gender × Stimuli	2	5.86	0.82	0.442	0.02

## Data Availability

All the data are available upon request from the corresponding author.

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
