# Peer review of "Distinct Emotional and Cardiac Responses to Audio Erotica between Genders"

_behavsci, 2023, doi:10.3390/bs13030273_

Round 1
Reviewer 1 Report
The paper describes an interesting study comparing the physiological (change in heart rate) and psychological (perception of pleasure and shame) effects of three different types of audio stimuli (neutral, happy, and erotic) on women and men. The main finding is that women reported higher levels of shame and perceived erotic audio stimuli as less pleasurable than happy ones, whereas men perceived erotic and happy audio stimuli as equally pleasurable. No significant differences were found in changes in heart rate.
Comments and notes
Line 165 - how do the authors explain the fact that they used audios that were not quite gender parity - e.g. moaning sounds were mainly women voices. Would not this imply that the stimuli were different for men and for women? This limitation is added to the list of study limitations, but the authors themselves write that "selection of stimuli could modulate sexual response" (line 303).
Line 166 - a brief explanation of why the (running?) water sound was added would be very helpful to the reader
Line 184 - were the ECG electrodes dry or wet (i.e., with electrolyte gel)? Were they attached with Velcro or self-adhesive? Were they wet and intended for single use?
Line 194 - the phrase "baseline corrected" is not entirely understandable - what do the authors mean? Did they subtract the HR value in the baseline from HR during the stimuli? Did they subtract the same baseline HR value from HR of all audio stimuli?
Lines 201 and 202 - Participant exclusion should be included in section 2.1 Participants.
Line 216 - Do the authors mean absolute differences in HR? A definition of "heart rate change" should be added.
Figure 3 A (HR change versus time) could be larger.
General note: One of the findings of this study was the change in heart rate value during exposure to three types of audio stimuli. These changes were modest, on the order of a few beats per minute. How can the authors convince the reader that the values are meaningful? Do they know the measurement uncertainty or even the measurement error of their ECG device (which was a fairly inexpensive device)? What if the uncertainty of the device is 3 bpm? How relevant are the results then?
Grammar/Typos
Line 37 - delete "these"
Line 47 - write "reported" instead of "evaluated"
Line 67 - replace "exposed" with "exposure"
Line 189 - use the correct font size
Author Response
Please find the revisions from the attached document.

Reviewer 2 Report
The authors are trying to determine the differences of emotional and cardiac responses to audio erotica among male and female respondents. I find this article is interesting for the readers, but I have a few comments as follows:
I’d change the font of “following consent, participants” line 130.
Out of curiosity, the Ethics committee has approved the use of porn website for the respondents. What is the basis?
Line 164-166- why do you use the sounds of water? What is the purpose?
Line 284-286, Cardiac data indicated that both genders showed comparable heart rate deceleration to erotica compared to neutral and happy. Can you explain why both genders would have cardiac deceleration to erotica stimuli.
I’d like to see more discussion on the practical aspect of their findings in behavioral or sexual therapy.
Author Response

(The authors gave the same response as above.)

Reviewer 3 Report
Dear authors,
Your work can be a basis for further study, using other sexual stimuli and erotic imagery.
I note a sparse case history and wondered if it could be expanded.
The other suggestion is to extend the study to TGNC or gender variant people.
My comments are in the article.
Best regards

Author Response

(The authors gave the same response as above.)
